# The Role of the Sdᵃ Carbohydrate Antigen and That of Its Cognate Glycosyltransferase B4GALNT2 in Health and Disease

Martina Duca, Nadia Malagolini and Fabio Dall'Olio *

Department of Medical and Surgical Sciences (DIMEC), University of Bologna, General Pathology Building, Via San Giacomo 14, 40126 Bologna, Italy; martina.duca3@unibo.it (M.D.); nadia.malagolini@unibo.it (N.M.)
* Correspondence: fabio.dallolio@unibo.it; Tel.: +39-0512094704

**Abstract:** The carbohydrate antigen Sdᵃ is expressed on the cells and secretions of the vast majority of Caucasians. The epitope is formed by a terminal GalNAc residue β4-linked to an α3-sialylated galactose. Different carbohydrate chains *N*- or *O*-linked to glycoproteins can be terminated by this epitope. The final step of Sdᵃ biosynthesis is catalyzed by the GalNAc transferase B4GALNT2. In this review, we discuss the multifaceted aspects of B4GALNT2/Sdᵃ in fertility and pregnancy, susceptibility to infectious diseases, cancer, chronic kidney diseases, and Duchenne muscular dystrophy. We show how multiple synthetic biology approaches have been adopted to investigate its role.

**Keywords:** Sdᵃ antigen; B4GALNT2; uromodulin; Tamm–Horsfall glycoproteins; influenza virus; Duchenne muscular dystrophy; synthetic biology

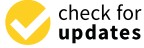

## 1. Introduction

About 96% of Caucasians display the carbohydrate antigen Sdᵃ on their erythrocytes, secretions [1], and a few organs [2]. This antigen, which behaves as a dominant genetic character, was discovered independently by two groups in 1967 [3,4]. Among the 4% of Sdᵃ-negative individuals, only a few contain "natural" anti- Sdᵃ antibodies in their serum [5]. This is in sharp contrast to the AB0 blood group system, in which all individuals form "natural" antibodies against the antigens they do not express. A stronger version of the Sdᵃ antigen, known as the Cad antigen, reacts more strongly with anti-Sdᵃ antibodies [6]. The structural and molecular differences between Cad and Sdᵃ are still not clear [5].

Basically, the structure of the Sdᵃ epitope is composed of an α2,3-sialylated galactose substituted by a β1,4-linked GalNAc [7]. However, this epitope can be found at the end of different sugar structures, including type 1 and type 2 lactosaminic chains (which decorate the *N*-linked, as well as *O*-linked, chains of glycoproteins and glycolipids), as well as core 1, core 2, and core 3 *O*-linked structures [8] and the glycolipid sialosylparagloboside [9] (Figure 1).

The UDP-GalNAc β1,4-N-acetylgalactosaminyltransferase 2 (B4GALNT2), encoded by the *B4GALNT2* gene, is the only enzyme able to catalyze the final step of Sdᵃ biosynthesis. This enzymatic activity was first detected in Guinea pig kidneys [10] and was found to require the presence of a sialic acid α2,3-linked to galactose in the acceptor. Mouse *B4galnt2* cDNA [11] and human *B4GALNT2* cDNA were successively cloned [12,13]. The human *B4GALNT2* gene maps onto 17q21.33; it is formed by at least 12 coding exons and generates transcripts diverging in their 5'- and 3'-UTRs (Figure 2). The presence of two alternative 5'-UTRs is particularly relevant. In fact, the alternative use of two distinct first exons, both provided with a translation start codon, results in two polypeptides with

different amino-terminal portions [12,13]. Transcripts containing either exon 1 long (1L) or exon 1 short (1S) encode for 566 aa or 506 aa long B4GALNT2 proteins. The long form is characterized by an unusually long cytoplasmic tail of 67 aa. Although both isoforms are mainly localized in the Golgi, the long form also displays post-Golgi vesicles and plasma membrane localization [14] (Figure 2). The B4GALNT2 protein contains two unconventional *N*-glycosylation sites. The first is occupied by a complex-type chain, which is necessary for stability, proper intracellular localization, and homodimer formation [15].

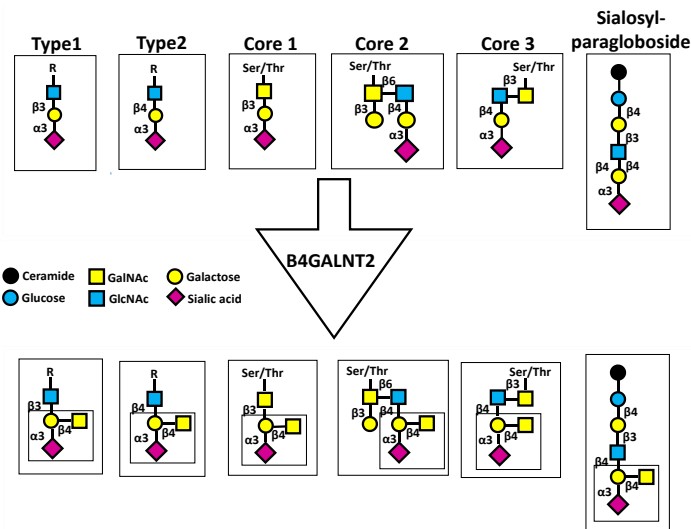

**Figure 1.** Structure of the Sd[a] antigen. Different carbohydrate structures of glycoproteins (first four structures) and the glycolipid sialosylparagloboside, all terminated by a sialic acid α3-linked to galactose (**upper panels**), can be substrates of B4GALNT2, which synthesizes the Sd[a] epitope (boxed) on these chains (**lower panels**). Type 1 and type 2 chains are commonly present in *N*-linked chains.

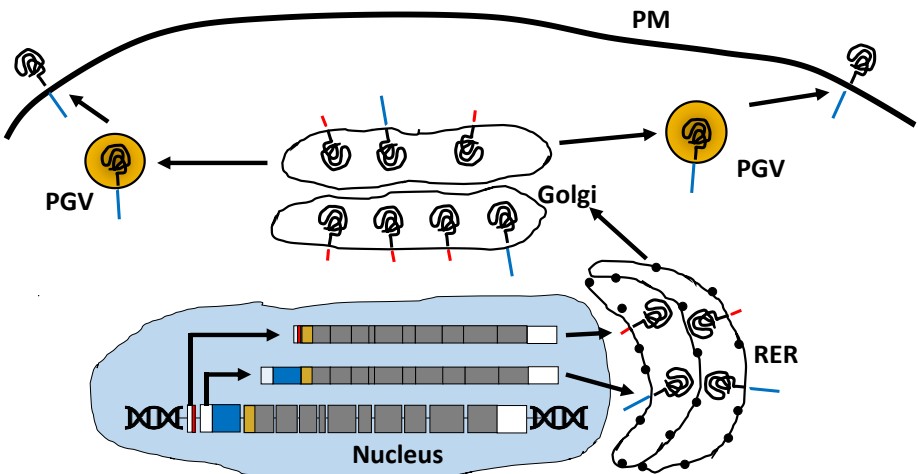

**Figure 2.** The *B4GALNT2* gene, transcripts, and proteins. *B4GALNT2*'s genomic organization is depicted, spanning at least 12 exons. The coding regions of the short and long forms of exon 1 are depicted in red and blue, respectively, while the 5'- and 3'UTRs are in white. The transmembrane domain is in yellow. Transcription of the two exons generates two polypeptides with different amino-terminal (cytoplasmic) domains, depicted in red and blue, respectively. After translation in the rough endoplasmic reticulum (RER), both are sorted to the Golgi apparatus, where they mainly localize. However, a portion of the long form also localizes in post-Golgi vesicles (PGVs) and the plasma membrane (PM). The cytoplasmic portions in red and blue are not drawn to scale.

The regulation of *B4GALNT2* expression is a complex issue. The genomic sequences upstream of the transcription start site(s) display the features of a CpG island and CpG shores. The methylation of these sequences can block *B4GALNT2* expression [16,17]. How-

ever, promoter methylation does not appear to behave as an all-or-none switch. In fact, data from The Cancer Genome Atlas (TCGA) reveal that the methylation status of specific sites in the promoter and inside the gene is more predictive of the expression level than methylation as a whole [18,19]. TCGA data have also revealed a potentially regulatory role of miR-204-5p [19]. Another crucial role in *B4GALNT2* regulation can be played by transcription factors. The transcription factors ETS1 and, to a lesser extent, SP1 are necessary for gene transcription, although their different expression does not appear to be responsible for B4GALNT2 modulation in pathological conditions, such as colon cancer (see below) [20].

Despite the fact that a small but significant percentage of the population is Sd$^a$/B4GALNT2-negative without overt phenotypic consequences, the Sd$^a$ antigen is involved in a broad variety of physio-pathological phenomena in humans and animals. The present review aims to focus on these physio-pathological phenomena, showing how synthetic biology was crucial for their investigation, mainly through the generation of unnatural cells and animals through genome alterations. A comprehensive history of the discovery of the Sd$^a$ antigen was published previously [8,21].

## 2. Why Are Some People Sd$^{a-}$?

The sequencing of the *B4GALNT2* gene from nine Sd$^{a-}$ individuals revealed a homozygous p.Cys466Arg substitution affecting the enzymatically active domain of the protein in six of them. In two other persons, a p.Gln436Arg mutation was associated either with a p.Arg523Trp substitution or with a splice site mutation [22]. Genetic engineering of an Sd$^{a-}$ cell line with the p.Cys466Arg mutant form failed to turn the cell line into an Sd$^{a+}$ status, confirming its lack of activity. Unexpectedly, both p.Gln436Arg and p.Arg523Trp mutant forms induced an Sd$^a$ expression level comparable with that of wild-type forms [23]. Another important question is in regard to the origin of the Cad status. Reasonably, it is possible that differences in the regulatory regions of the B4GALNT2 gene allow for a stronger expression in Cad individuals. However, neither the coding sequence nor the genomic region 2000 bp upstream of their *B4GALNT2* gene revealed common alterations, potentially accounting for the Cad status [23]. In conclusion, some of the Sd$^{a-}$ phenotypes are due to mutations in the coding sequence of B4GALNT2, but the origin of other cases remains obscure [5].

## 3. What Are the Evolutionary Forces Driving the Selection of Sd$^a$ Phenotypes?

The answer to this question involves at least two aspects: (1) resistance to microorganism infection and (2) the regulation of prolificacy.

### 3.1. How B4GALNT2/Sd$^a$ Regulates Microbe Infections

#### 3.1.1. Viral Infections

Viruses often exploit sialylated carbohydrate structures on the host cell surface as receptors for their sugar-binding proteins [24]. A very good example of these mechanisms is represented by influenza viruses, whose lectin, called hemagglutinin (H), binds to the sialic acid residues on host cell glycoconjugates. Various influenza virus strains and their tropism for species and tissues are distinguished according to their different H molecules (H1, H2, etc...) with different specificities for $\alpha$2,3- or $\alpha$2,6-linked sialic acids [25]. The CRISPR synergistic activation mediator (CRISPR SAM) is a synthetic-biology-based adaptation of CRISPR/Cas9 technology, which allows genome-wide gene overexpression screens to be performed by recruiting transcriptional activators. With this approach, B4GALNT2 was identified as the major factor whose overexpression inhibits the binding of avian influenza

viruses, including the α2,3 sialic acid-specific H5, H9, and H7 strains, previously found to cause disease in humans [26]. However, owing to the strict specificity of B4GALNT2 to α2,3-sialylated chains, this inhibitory activity is not expected to involve hemagglutinins specific to α2,6-sialylated glycans. This notion has been experimentally confirmed by the reduced entry and infection of influenza virus strains expressing hemagglutinins specific to α2,3-sialylated chains in MDCK cells engineered to express *B4GALNT2* [27]. In chicken fibroblasts overexpressing human *B4GALNT2,* the entry and infectivity of avian influenza virus and Newcastle disease virus were reduced. Both virus types mainly expressed receptors for α2,3-linked sialic acid [28].

### 3.1.2. Bacterial Infections

The role of B4GALNT2/Sd[a] in bacterial interactions is complex (for an extensive review, see [29]). One study reported differences in the intestinal microbiota composition between wt and *B4galnt2* KO mice [30]. The distribution of phyla and individual bacterial species varies among gastrointestinal mucosa tracts. The most remarkable differences between wt and *B4galnt2* KO mice include a reduced proportion of Proteobacteria, a parallel increase in Bacteroidetes, and the virtual absence of *Helicobacter spp.* in the intestines of *B4galnt2* KO mice [30]. This suggests that the absence of Sd[a] provides a fitness advantage towards bacteria requiring terminal β4-linked GalNAc for their adhesion. Consistently, one study showed reduced effects of *Salmonella typhymurium* infection in mice deficient in intestinal *B4galnt2* expression because of the reduced production of inflammatory cytokines and immune cell infiltration [31]. The presence of intestinal B4galnt2 is also associated with increased susceptibility to *Morganella morganii* [32] but decreased susceptibility to *Citrobacter rodentium* infection, a murine model pathogen for human enteropathogenic *Escherichia coli* [33]. B4galnt2/Sd[a] can be naturally absent from the intestinal epithelium in mouse laboratory strains and wild mouse populations [34]. In fact, B4galnt2 can be alternatively expressed in the intestine or in the endothelial cells as determined by a polymorphic region located 30 kb upstream of the *B4galnt2* gene. This region is present in two allelic forms [34]: the C57 allele (present in the C57BL6/J mouse strain) is responsible for its intestinal expression, while the RIII allele (present in the RIIIS/J mouse strain) determines its endothelial expression [35]. Both alleles may also be present in wild mouse populations [36], together with a third allele dictating neither endothelial nor epithelial localization [37]. When the endothelial cells express B4galnt2, the Von Willebrand clotting factor (vWf) becomes decorated by the Sd[a] antigen [36]. This leads to its recognition and quick removal from circulation by the asialo glycoprotein receptor (a liver lectin that removes glycoproteins terminating with galactose or GalNAc from the blood circulation), thus resulting in a bleeding disorder [35–37]. This obviously detrimental condition is also observed in wild mice, suggesting a putative advantage of B4galnt2 loss in the intestinal tissues, probably due to reduced pathogen susceptibility. In the kidneys, the Sd[a] antigen is carried by the *N*-linked chains of uromodulin (also known as Tamm–Horsfall glycoprotein) [7]. It has been proposed that the presence of β4-linked GalNAc hinders the binding of pyelonephritogenic *E. coli* strains through their S-fimbriae specific to Siaα3Galβ4GlcNAc [38]. It may be hypothesized that the opposite effect that B4galnt2 expression has on the susceptibility to different bacterial species in mice is due to the specificity of the microbial adhesion molecules. If those specific to terminal galactose are prevalent on those specific to Siaα2,3Gal chains, then the Sd[a] antigen could help infection. Otherwise, it may be protective. The crucial role of B4GALNT2/Sd[a] in protection against bacteria is supported by the finding that they are expressed only after birth upon bacterial colonization in both rats [39] and humans [40].

### 3.1.3. Worm Infections

The gastrointestinal round worm *Nippostrongylus brasiliensis* infects rodents. During its infection, mucin glycosylation is altered by the induction of *B4galnt2*, leading to the biosynthesis of *O*-linked oligosaccharides carrying the Sd$^a$ epitope [41–43]. This provides an example of how besides bacteria, metazoan pathogens can also shape intestinal glycosylation.

### 3.2. *How the B4GALNT2/Sd$^a$ Antigen Regulates Reproduction*

In animals, B4GALNT2/Sd$^a$ plays multiple roles associated with both the pre- and post-fertilization steps of reproduction. Here, we discuss its role in gamete formation, implantation, and regulation of the mother's immune response.

### 3.2.1. Gametes

The Sd$^a$ antigen is expressed by both the male and female gametes of various animals [44] and, in particular, in the murine zona pellucida of the oocytes [45]. The involvement of B4GALNT2 in the prolificacy of certain breeds of sheep is indicated by the following findings. In some breeds, ovulation rates and fertility are associated with the genetic locus FecL, in which Fec(L) is the high-prolificacy allele [46,47]. Of the genes contained in the FecL locus, *B4GALNT2* is that responsible for increased prolificacy because it is ectopically overexpressed (about 1000-fold) in granulosa cells in Fec(L)-carrier animals only [46]. As a consequence, the Sd$^a$ antigen becomes expressed by some glycoproteins, including inhibin, an important hormone regulating ovarian function [46]. However, in breeds of Chinese origin, B4GALNT2 affects prolificacy through quite different mechanisms [48,49].

### 3.2.2. Implantation

In mouse uterine tissues, *B4galnt2* gene expression is stimulated by placental progesterone production [50]. *B4galnt2's* down-regulation by siRNA treatment results in a reduced number of implanted embryos [50]. Moreover, the attachment of blastocysts to endometrial cells in vitro can be inhibited either by antibodies against *B4galnt2* or by lectins recognizing the Sd$^a$ epitope [51]. A recent paper investigated the relationships among B4galnt2 expression, implantation, and preeclampsia [52]. Preeclampsia, a major cause of maternal and neonatal morbidity and mortality, is defined as a complication of pregnancy developing after the 20th week of gestation, characterized by hypertension with or without edema and proteinuria. Although the etiopathogenesis of preeclampsia still remains unclear, a crucial role is played by an abnormally implanted placenta, which results in poor uterine and placental perfusion. Proper placental implantation requires adequate trophoblast invasion of the uterine epithelium, a process associated with spiral artery remodeling. In murine models, a lack of maternal galectin-1 causes preeclampsia because of an insufficient trophoblast invasion ability [52]. Galectin-1 stimulates the expression of *B4galnt2* and consequently of Sd$^a$-capped *N*-glycans by trophoblast cells, which are necessary for invasion [52].

### 3.2.3. Regulation of the Mother's Immune Response

Glycodelin is a glycoprotein present in four differentially glycosylated forms. Three are from the female genital tract, and one is from sperm. A portion of the *N*-linked chains of the three female glycodelins is terminated by the Sd$^a$ antigen [53]. Glycodelin A is present in the amniotic fluid and contributes to protecting the fetus from rejection by the mother's immune system by skewing T-cell differentiation toward the Th-2 phenotype and the inhibition of NK activity [53]. This immunosuppressive activity is largely due to $\alpha$2,6-sialylated glycodelin A. However, if $\alpha$2,6-sialylation decreases in favor of increased

Sd$^a$ expression, as occurs in gestational diabetes mellitus, the immunosuppressive activity of glycodein A is reduced [54].

In conclusion, data from sheep and mice indicate the positive role of B4GALNT2 in promoting prolificacy, while data from different mouse strains indicate either a positive or a negative effect of intestinal B4GALNT2 expression. In the vast majority of humans, B4GALNT2 is expressed in the blood and large intestine, suggesting a possible positive role.

## 4. How B4GALNT2/Sd$^a$ Plays a Role in Cancer

B4GALNT2/Sd$^a$ display a strong dependence on neoplastic transformation. Investigations on a limited number of colon cancer cases have previously shown the dramatic down-regulation of B4GALNT2 enzyme activity [55], B4GALNT2 mRNA [56], and Sd$^a$ antigen mRNA [57,58]. The availability of large public databases, such as TCGA, reporting the clinical and molecular data of hundreds of patients has confirmed the marked reduction in *B4GALNT2* mRNA in colorectal cancer cases [19,59]. In most patients, *B4GALNT2* is completely switched off. However, patients retaining a nearly normal expression level display a much longer overall survival [19,59]. These patients display high levels of genes related to normal colon functions, such as mucus secretion, ion transport, and proper glycosylation, while genes associated with tumor progression, such as *WIF1* and *TSIX,* are poorly expressed [19]. In ulcerative colitis, a pre-neoplastic intestinal inflammatory condition, *B4GALNT2* is also transcriptionally modulated [60]. In the colon, the biosynthesis of Sd$^a$ is strictly connected to that of the sialyl Lewis x (sLex) [Sia$\alpha$2,3Gal$\beta$1,4(Fuc$\alpha$1,3)GlcNAc] antigen [61] (Figure 3), which acts as a ligand for the cell adhesion molecules of the selectin family, playing a physiological role in leukocyte extravasation and a pathological role in metastasis formation [62].

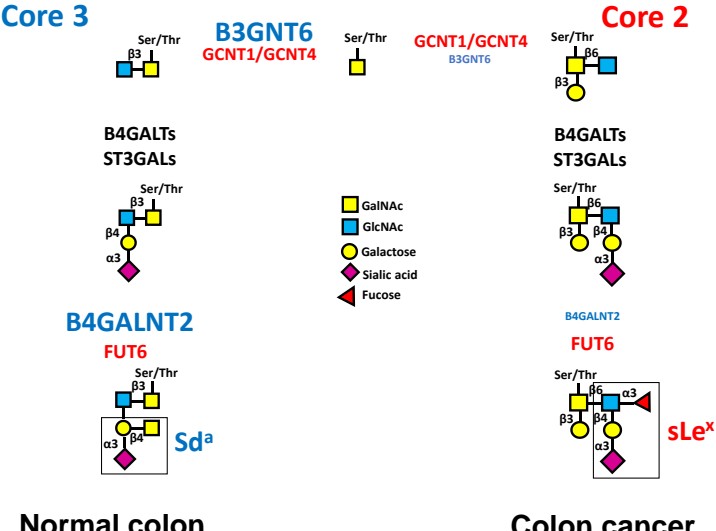

**Figure 3.** Alternative biosynthesis of Sd$^a$ and sLe$^x$ in the normal colon and colon cancer. In the normal colon, the biosynthesis of the core 3 structure is largely predominant over that of core 2 because of the high level of B3GNT6, while the high level of B4GALNT2 prevents the biosynthesis of sLe$^x$. In colon cancer, the down-regulation of B3GNT6 leads to the replacement of core 3 with core 2 structures, while the down-regulation of B4GALNT2 leads to sLe$^x$ expression. GCNT1 and GCNT4 are core 2 synthases. The size of the lettering roughly represents the expression level of the enzyme.

Due to steric hindrance, the biosynthesis of sLe$^x$ and Sd$^a$ antigens is believed to be mutually exclusive. In fact, GalNAc (Sd$^a$) and Fuc (sLe$^x$) are added by the respective glycosyltransferases to two adjacent sugars (Figure 3). The Sd$^a$ epitope, but not sLe$^x$, is expressed by mucins from the normal colon, while GalNAc $\beta$4-linked to galactose and Fuc $\alpha$3-linked to GlcNAc in adjacent positions has never been detected [63]. The transition

from a normal colon to cancer is accompanied by a switch from *O*-linked chains with core 3 structures, which are often decorated by Sd[a], to those with core 2 structures, which are often terminated by sLe[x] antigens [64] (Figure 3). The core 3 to core 2 transition is mainly due to the down-regulation of core 3 biosynthesis rather than to the up-regulation of core 2 biosynthesis [64] (Figure 3). In addition, B4GALNT2 levels play a role in maintaining low sLe[x] biosynthesis in the normal colon by competing with FUT6 for the same substrate acceptor(s) [65]. Thus, the switch from core 3/Sd[a] in the normal colon to core 2/sLe[x] in colon cancer is due to the down-regulation of B3GNT6 and B4GALNT2 rather than to the up-regulation of core 2/sLe[x] synthases and FUT6 in cancer (Figure 3). Interestingly, the core 3 synthase B3GNT6 and B4GALNT2 display a roughly proportional positive relationship with B4GALNT2 in the colon adenocarcinoma (COAD) TCGA cohort [19].

In the stomach, the level of *B4GALNT2* expression is about 50-fold lower than that in the colon, while in the near totality of gastric cancer samples, it is virtually switched off [18]. Consequently, the Sd[a] antigen is lost in gastric cancer [66].

Even if the kidney is a major site of B4GALNT2/Sd[a] expression, no data have been reported so far on its modulation in malignancy. TCGA reports two kidney cancer cohorts: kidney renal clear cell carcinoma (KIRC), which accounts for 70–80% of cases, and kidney renal papillary carcinoma (KIRP), accounting for 10–15%. In both cohorts, the B4GALNT2 expression was markedly reduced in cancer samples (Figure 4A,B,D,E), although to a variable degree. In KIRC patients, the relationship of B4GALNT2 with survival is complex and non-significant (Figure 4C). In fact, in the first 1500 days, high B4GALNT2 expressers displayed better survival, while after 1500 days, the opposite occurred. We observed that the putative tumor suppressor gene *HEPACAM2* is expressed only by high *B4GALNT2* expressers. Like in colon cancer (see above), *TSIX* was strongly down-regulated in high B4GALNT2 expressers. In KIRP, the association between high B4GALNT2 expression and longer survival is highly significant (Figure 4F).

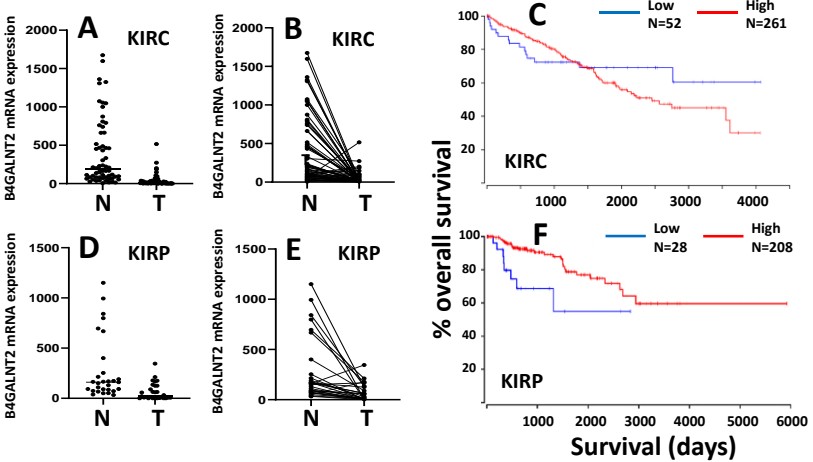

**Figure 4.** B4GALNT2 in kidney cancers. In both the KIRC (**A**,**B**) and KIRP (**D**,**E**) cohorts, B4GALNT2 was strongly down-regulated in tumor tissue (T) compared to normal tissue (N). The Kaplan–Meier survival curves (**C**,**F**) indicate that in KIRP, high expressers (red line) display a better prognosis ($p = 0.005$), while in the KIRC cohort, the relationship is more complex. The data were downloaded from TCGA as described [59]. The Kaplan–Meier curves were obtained as described [67]. For KIRC, the low (blue line) and high (red line) expresser percentiles were 10 and 50, while for KIRP, they were 10 and 73%, respectively.

The involvement of B4GALNT2/Sd[a] in breast cancer is suggested by TCGA data, showing that in normal breast tissue, *B4GALNT2* is virtually not expressed. However, in a minority of breast cancer cases, it reaches a relevant level, suggesting ectopic expression of the gene. In contrast with colon cancer, patients expressing high *B4GALNT2* display

a shorter overall survival and a gene expression profile closely associated with malignancy [18]. The overexpression of the *KRT20* gene, encoding cytokeratin 20, which is about 500-fold higher in high compared to low expressers, is particularly relevant.

An association does not necessarily imply a causal relationship. To establish the existence of a causal nexus between B4GALNT2 expression and malignancy, synthetic biology has been widely used to force Sd$^a$ antigen biosynthesis through gene transfer experiments. Previous experiments in colon cancer have shown that forced *B4GALNT2* expression results in strong inhibition of the sLe$^x$ antigen [57,68] and a reduction in the metastatic potential of transfected cells [68,69]. This finding raises a key question: is this effect attributable to the inhibition of the sLe$^x$ antigen or is it independent? To answer this question, the two colon cancer cell lines SW480 and SW620 (the former derived from a primary tumor and the latter from a metastasis of the same patient), originally devoid of both the sLe$^x$ and Sd$^a$ antigens, were forced to express either Sd$^a$ through *B4GALNT2* transfection or sLe$^x$ through transfection with the main sLe$^x$ synthase *FUT6* [70,71]. Consistent with previous data obtained using the LS174T cell line [59], in SW480 and SW620, B4GALNT2 expression reduced stemness [71]. Importantly, this model showed that the phenotypic effects were oriented toward reduced malignancy, independent of sLe$^x$ inhibition [71]. Consistent with the TCGA data, experimental studies with triple-negative breast cancer cell lines have shown a positive correlation between high B4GALNT2 and malignancy [72,73]. In particular, it has been shown that the B4GALNT2 protein is able to interact with the proteins of the major histocompatibility complex (HLA-B) [73]. Thus, in both colon- and breast cancer, experimental data show that the level of B4GALNT2 expression strongly affects the behavior of cancer cells.

## 5. How B4GALNT2/Sd$^a$ Could Cure Duchenne Muscular Dystrophy

Duchenne muscular dystrophy (DMD) is an extremely severe condition caused by the absence of the cytoplasmic protein dystrophin. The biological role of dystrophin is to connect the cytoskeleton of the muscle cells with membrane β-dystroglycan, which, in turn, is connected to α-dystroglycan and laminin in the extracellular matrix [74]. The use of synthetic biology approaches has been crucial to investigating the role of B4GALNT2/Sd$^a$ in DMD. Mouse models of DMD were established through genome modification. Their successive modification through *B4galnt2* gene transfer was then shown to protect the skeletal muscle cells from injury [75–78]. The mechanisms underlying this phenomenon are very complex and have only partially been elucidated [79–81]. *B4galnt2* gene delivery through a viral vector in a dog model of DMD was revealed to be safe and able to induce B4galnt2 expression, although with little to no improvement in the pathology [82]. The administration of the viral vector to two DMD-affected boys resulted in functional improvements only in the younger patient treated with a higher dose of the *B4GALNT2* vector [83].

## 6. How B4GALNT2/Sd$^a$ May Play a Role in Kidney Disease

Uromodulin/Tamm–Horsfall glycoprotein (see above), the product of the *UMOD* gene, represents the major urinary protein of healthy individuals and is a major carrier of the Sd$^a$ antigen [38]. Genome-wide association studies (GWASs) have identified some polymorphisms in the *UMOD* gene as risk factors for chronic kidney disease. These genetic variants are associated with increased transcription of the *UMOD* gene in the kidneys and of the uromodulin level in the urine and serum [84]. In the search for other genetic loci whose polymorphisms affect uromodulin levels, Li et al. [84] found that the p.Cys466Arg-inactive variant of *B4GALNT2* (see above) is also associated with increased uromodulin levels in the serum. It is conceivable that the lack of terminal GalNAc in the uromodulin of p.Cys466Arg

individuals reduces its uptake by the asialo glycoprotein receptor (see above). However, a demonstration of the higher susceptibility of Sd$^{a-}$ individuals to chronic kidney disease is still lacking.

## 7. How B4GALNT2/Sd$^a$ Affects Xenotransplantation

The availability of organs for transplantation from deceased persons represents a strongly limiting factor in modern surgery. The use of pig organs may be a possible solution but is strongly limited by the occurrence of hyperacute rejection. This reaction takes place when the recipient contains preformed antibodies against the surface antigens of the donor organ, which leads to complement-mediated rejection in a very short time [85]. In pig organs, there are at least three carbohydrate antigens that potentially trigger hyperacute rejection by human blood. One of these is the so-called "Galili antigen", consisting of a terminal $\alpha$1,3-linked galactose residue, whose addition is mediated by the $\alpha$-galactosyltransferase encoded by the *GGTA1* gene [86]. During evolution, *GGTA1* was inactivated in a precursor of humans and Old World primates (gorillas, bonobos, chimpanzees, etc.) [87]. Consequently, these species do not express the Galili antigen on their cells, while a significant percentage of their circulating antibodies is able to react with it [87]. This is probably due to cross-reactivity with microorganism antigens and a lack of tolerance of the human immune system to this antigen. A second type of non-human carbohydrate is represented by glycoconjugates terminating with *N*-glycolylneuraminic acid [87]. This type of sialic acid is present in animals but not in human glycoconjugates because the *CMAH* gene encoding a CMP-N-acetylneuraminic acid hydroxylase was inactivated after the divergence of humans from Old World apes [88]. Surprisingly, the third is the Sd$^a$ antigen encoded by porcine *B4GALNT2* [89,90]. In fact, human cells forced to express porcine *B4GALNT2* display increased complement-mediated lysis with serum from primates pre-immunized with pig organs [91]. The construction of different strains of pigs lacking *CMAH/GGTA1/B4GALNT2* in different combinations, pursued by different labs, provides a good example of system biology being applied to transplantation science. Blood cells from *CMAH/GGTA1/B4GALNT2* triple-KO pigs display reduced reactivity with human plasma compared to cells from *CMAH/GGTA1* double-KO pigs [92–95]. It is not clear why the human immune system is not tolerant toward a self-antigen when it is synthesized by pig B4GALNT2. Whatever the reason, the presence of Sd$^a$ antigens on pig cells is an obstacle that must be removed before pig-to-human transplantation can be considered.

## 8. Conclusions

Even though a small but significant fraction of the human population lacks the Sd$^a$ antigen on its cells and secretions without obvious pathological consequences, many lines of evidence indicate that this antigen may be involved in determining fertility and susceptibility to microbial pathogens or kidney diseases. In addition, the absence or presence of Sd$^a$ may be crucial to the outcome of certain malignancies. A large body of the experimental data on the physio-pathological role of B4GALNT2/Sd$^a$ has been obtained through an analysis of unnatural cells or animals generated using genome manipulation. A more in-depth comprehension of these phenomena and their exploitation for therapeutic purposes will require a more extensive and refined use of synthetic biology approaches.

**Author Contributions:** All authors contributed to the writing—review and editing. All authors have read and agreed to the published version of the manuscript.

**Funding:** This research received no external funding.

**Conflicts of Interest:** The authors declare no conflicts of interest.

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
