# Peer review of "The Role of the Sda Carbohydrate Antigen and That of Its Cognate Glycosyltransferase B4GALNT2 in Health and Disease"

_2674-0583, doi:10.3390/synbio3010006_

Round 1

Reviewer 1 Report

Comments and Suggestions for Authors

The review article entitled “The Sda carbohydrate antigen and its biosynthetic enzyme B4GALNT2 in health and disease” examines the carbohydrate antigen Sda and its biosynthesis enzyme B4GALNT2, analyzing their biology and their importance in various physiological and pathological conditions. The Sda antigen, expressed in cells and secretions of the majority of Caucasians, is formed by a GalNAc residue linked to an α3-sialylated galactose. The final step of Sda biosynthesis is catalyzed by GalNAc transporter, B4GALNT2. The article explores the contribution of B4GALNT2/Sda to fertility, pregnancy, susceptibility to infections, chronic kidney disease, Duchenne dystrophy, and cancer.
1. The review article attempts to answer how Sda antigen and B4GALNT2 enzyme are involved in various physiological and pathological conditions in humans and animals.
2. The article is considered original as it comprehensively reviews the role of B4GALNT2/Sda in different systems of the human body and in various diseases. It addresses the gap in understanding the multiple impact of this antigen and enzyme on various biological processes and diseases.
3. Compared to other published reviews, the article offers a comprehensive analysis of the Sda antigen and the B4GALNT2 enzyme, discussing their involvement in many diseases and their impact on physiology and pathology.
4. This manuscript has been re-reviewed as biomolecules-2915476, and the authors have made all requested improvements in terms of figure quality and upgrading the quality of the English language.
5. The conclusions of the article are broadly consistent with the evidence and arguments presented. The main questions were addressed through a variety of experiments and data analysis.
6. References appear appropriate and cover a wide range of existing literature.
7. Tables and figures are well designed and help to understand the information. Data quality is high and well documented.
In my opinion the manuscript could be published in its present form.

Author Response

  1. The review article attempts to answer how Sda antigen and B4GALNT2 enzyme are involved in various physiological and pathological conditions in humans and animals.1. The review article attempts to answer how Sda antigen and B4GALNT2 enzyme are involved in various physiological and pathological conditions in humans and animals. Yes, this was the aim of our work.
  2. The article is considered original as it comprehensively reviews the role of B4GALNT2/Sda in different systems of the human body and in various diseases. It addresses the gap in understanding the multiple impact of this antigen and enzyme on various biological processes and diseases. Thanks.

  3. . Compared to other published reviews, the article offers a comprehensive analysis of the Sda antigen and the B4GALNT2 enzyme, discussing their involvement in many diseases and their impact on physiology and pathology. Thanks.

  4. This manuscript has been re-reviewed as biomolecules-2915476, and the authors have made all requested improvements in terms of figure quality and upgrading the quality of the English language. Yes, correct.

  5. The conclusions of the article are broadly consistent with the evidence and arguments presented. The main questions were addressed through a variety of experiments and data analysis. Thanks.

  6. References appear appropriate and cover a wide range of existing literature. Thanks.
  7. Tables and figures are well designed and help to understand the information. Data quality is high and well documented.
    In my opinion the manuscript could be published in its present form. Thanks.

Reviewer 2 Report

Comments and Suggestions for Authors

Short summary of article

In the current review the authors explain the role of the carbohydrate antigen Sd (GalNac linked to the a sialylated galactose) on physio pathological phenomena

minor comment :

Line 24: “was discovered independently by two groups” would be nice to read when was discovered

Figure 1 “Structure of the Sda antigen” here you depict only O-glycosylation, as you describe Sd antigen in N glycosylation (line 205) would be interesting to observe some example of the N-glycosilation structures.

It would also be interesting to show schematically the expression of the Sd antigen in the different diseases

Author Response

Comment 1. Line 24: “was discovered independently by two groups” would be nice to read when was discovered. The year of discovery was 1967. This information has been added in line24.

Comment 2 Figure 1 “Structure of the Sda antigen” here you depict only O-glycosylation, as you describe Sd antigen in N glycosylation (line 205) would be interesting to observe some example of the N-glycosilation structures. In general, on N-linked chains the Sda antigen is carried by type 1 or type 2 (mainly type 2) lactosaminic structures, like those depicted in Fig. 1. This information has been added in the legend of Fig. 1 and in lines 32-33 of the new text.

Comment 3. It would also be interesting to show schematically the expression of the Sd antigen in the different diseases. This is a very hard task. In fact, while the information on B4GALNT2 expression in cancers is abundant thanks to TCGA, the information on the Sda antigen are not so abundant in different pathologies because they require structural analysis which so far have been performed mainly in colon.